# Evaluating the Histologic Grade of Digital Squamous Cell Carcinomas in Dogs with Dark and Light Haircoat—A Comparative Study of the Invasive Front and Tumor Cell Budding Systems

**DOI:** 10.3390/vetsci8010003

**Published:** 2020-12-30

**Authors:** Argiñe Cerezo-Echevarria, Julia M. Grassinger, Christoph Beitzinger, Robert Klopfleisch, Heike Aupperle-Lellbach

**Affiliations:** 1Pathology Department, LABOKLIN GmbH & Co. KG, 97688 Bad Kissingen, Germany; grassinger@laboklin.com (J.M.G.); beitzinger@laboklin.com (C.B.); aupperle@laboklin.com (H.A.-L.); 2Institute of Veterinary Pathology, Freie Universität Berlin, 14163 Berlin, Germany; robert.klopfleisch@fu-berlin.de

**Keywords:** digital squamous cell carcinoma, canine, cancer, tumor budding, digital, toe, squamous cell carcinoma, grading, haircoat color, genotype

## Abstract

**Simple Summary:**

This study compares two different adapted grading systems for Canine digital squamous cells carcinomas, taking into account the animals’ haircoat color and focusing on the tumor’s invasive front. In general, dark-haired breeds develop more poorly differentiated DSCC than their light-haired counterparts. Additionally, both grading systems challenged are in agreement when grading well differentiated CDSCC in both populations but are discordant when assessing tumors with poorly differentiated features. To our knowledge, this is the first study comparing CDSCC in dogs by two histological grading systems, taking into account their phenotypical and presumed genotypical haircoat color and demonstrating that digital squamous carcinomas are not only more common in dark-haired dogs, but potentially more aggressive.

**Abstract:**

Canine digital squamous cell carcinomas (CDSCC) are particularly aggressive when compared to their occurrence in other locations. Although these neoplasms are more frequently seen in dark-haired dogs, such as Giant Schnauzers, there are no data checking whether these tumors are histologically different between breeds. We histologically evaluated DSCC from 94 dogs. These were divided into two groups, namely, (1) dark-haired (N = 76) and (2) light-haired breeds (N = 18), further subdividing Group 1 into three subgroups, (1a) black breeds (*n* = 11), (1b) Schnauzers (*n* = 34) and (1c) black & tan breeds (*n* = 31). Adaptations from two different squamous cell carcinomas grading schemes from human and veterinary literature were used. Both systems showed significant differences when compared to Groups 1 and 2 in terms of final grade, invasive front keratinization, degree of invasion, nuclear pleomorphism, tumor cell budding, smallest tumor nest size and amount of tumor stroma. Group 2 was consistently better differentiated CDSCC than Group 1. However, there were no significant differences among the dark-haired breeds in any of the features evaluated. This study represents the first attempt to grade CDSCC while taking into account both phenotypical and presumptive genotypical haircoat color. In conclusion, CDSCC are not only more common in dark-haired dogs, they are also histologically more aggressive.

## 1. Introduction

Squamous cell carcinoma (SCC) is a fairly common, locally invasive and destructive neoplasm arising from the epithelium with keratinocyte differentiation. This neoplasia is known to metastasize in late stages of the disease, being variably prone to it depending on its anatomical location [1,2]. Some of them, especially those arising from the nail bed or digits in dogs, are known to be particularly aggressive [1]. A number of factors are associated with the development of these tumors in veterinary species, including papilloma-induced neoplasms and chronic ultraviolet (UV)-damage, especially in poorly haired, light skin [1]. In proposed nondigital, UV-induced SCC, precancerous, actinic changes are often reported within the neighboring tissue [1,3]. This type of SCC is associated with slower progression and overall longer survival [3]. 

Curiously enough, canine subungual/digital squamous cell carcinomas (CDSCC) are most commonly seen in dark, large breeds such as Giant Schnauzers, black Labrador Retrievers and standard Poodles [1,4]. Papilloma virus was thought to be associated with CDSCC, but was not demonstrated by PCR positivity in affected digits [5]. 

Squamous cell carcinoma gradings are more widely explored in human medicine, conducting different grading schemes based on their location, such as the esophagus [6], uterine cervix [7], lung [8], larynx and hypopharynx [9], among others [10]. In veterinary medicine, a standardized grading system is not well characterized given its unclear prognostic value to date [11]. This is mainly because, in the toe, complete digit amputation is considered the only treatment option and, often curative [12]. Currently, the most used grading system is Broder’s system [13] in which canine SCC is characterized as “well differentiated/I”, “moderately differentiated/II and III” and “poorly differentiated/IV”. The grading is based on its general morphologic features and its resemblance to normal squamous epithelium [14]. This, however, often ends up being the pathologist’s subjective assessment (especially for tumors of grade II and III) and, with no proven prognostic correlation, discourages SCC subtyping in a diagnostic setting. Also, newer canine SCC gradings often focus on the oral cavity due to its more malignant behavior [15]. 

Recently, more research was conducted on the tumor invasive front and epithelial-mesenchymal transition, both in human [7,9,15,16] and veterinary medicine [2]. These features are associated with the pattern of invasion and, therefore, malignancy. The invasive front, as its name infers, is the tumor–host interface, in which neoplastic cells invade the surrounding stroma, spreading and infiltrating. The reason for studying invasive fronts in SCC is that within the same tumor, different grades of differentiation can be found but the invasive front consistently has more malignant features [2,11]. This suggests that it may be imperative for neoplastic infiltration and expansion. Features that are associated with more malignant behavior include tumor budding, which are small aggregates (less than five cells) or single tumor cells that detach from the primary tumor and invade into the surrounding stroma [2,7,9,10,14,15,16]. 

While CDSCC is the most common neoplasia in the canine digit (up to 47.4% of all malignant digital tumors) [17], there is not much literature available [11]. Canine squamous cell carcinoma, particularly that developing in dark-haired breeds, garnered special interest, suggesting an underlying genetic predisposition [11]. Different canine haircoat colors and distribution are due to eumelanin (dark) and pheomelanin (light) pigments, which are also responsible for the claw coloration. Dark-haired animals, such as Giant Schnauzers, black Labrador Retrievers and Poodles, normally have concurrent dark claws. However, light-haired dogs with recessive genotype e/e on the E-locus do not incorporate eumelanine into their hair or claws, hence the light appearance [18]. This presumably important gene locus is homozygous recessive (e/e) for some breeds (e.g., Golden Retriever), while completely absent and therefore homozygous wildtype (E/E) in others (e.g., black Russian Terrier). Interestingly enough, Poodles, Labrador Retrievers and some Schnauzer variants have individuals with either homozygous states (E/E or e/e) in their breed. 

The KIT ligand (KITLG) locus, associated with postnatal cutaneous melanogenesis and follicular epithelial melanocyte terminal differentiation (among other functions), was shown to play a significant role in canine haircoat pigmentation [19]. One study identified a copy number variant at the KITLG locus in animals with CDSCC, which may predispose them to develop this neoplasia [20]. 

The objective of this study was (1) to compare two adapted grading schemes from both human and veterinary medicine for canine digital squamous cell carcinoma, and (2) to evaluate if there are significant characteristic disparities between light and dark coated dogs, based on the grading schemes discussed and taking into account their phenotypical haircoat color.

## 2. Materials and Methods 

### 2.1. Samples

Out of the 2983 toes submitted between 2014 and 2019 for routine diagnostics to the pathology department of LABOKLIN GmbH and Co. KG, 53% (*n* = 1576) of them contained a tumor, of which 49% (*n* = 771) were CDSCC. Out of these, 39% were found in Schnauzer breeds and only 2.5% in Golden Retriever (unpublished data). Given that these samples were from dogs subjected to regular routine diagnostics and not sampled for pure research purposes, an Ethical Committee approval was not necessary before undertaking the research. 

Histological samples of CDSCC from 94 dogs with a clear neoplastic invasive front, available breed and haircoat color were included in this study. All CDSCC from animals of unclear haircoat color, unknown breed or only including neoplastic fragments with no clear invasive front were excluded.

Dog grouping for this study followed the main color of their fur and claws (light/dark), as well as the presumed underlying genetics for the color loci A, K and E, which are most important in the distribution of light and dark pigment in hairs and claws. 

In Group 1, the phenotypically “dark-haired breeds”, composed of 76 dogs, was further divided into three subgroups: Group 1a (*n* = 11) was made by presumed genetically entirely black breeds (presumed KB/KB) including seven Russian Terriers, two black Briard, one black Giant Poodle, and one black Labrador Retriever. Group 1b (*n* = 34) (presumed KB/KB and KB/KY) were represented by 27 Giant Schnauzers and seven black standard Schnauzers. Group 1c (*n* = 31) consisted of genetically black & tan (presumed KY/KY + at/*) breeds, represented by 22 Rottweilers and 10 Gordon Setters. Group 2 (*n* = 18) were the light-colored breeds (presumed e/e) including 15 Golden Retrievers and three West Highland White Terriers. No genetic testing was performed to corroborate the presumed genotype in any of the groups. 

The ages of the dogs ranged from 6 to 14 years, with a median of 10 years, as one animal’s age was unknown. Sex was either unknown (*n* = 7), female intact (*n* = 18), female spayed (*n* = 18), male (*n* = 34) and male castrated (*n* = 18). Limb and toe affected, when available, was noted. Signalment is summarized in Table 1 and individual cases with more detailed information can be seen in Appendix A.

All digital samples were fixed in 10% phosphate-buffered formalin, routinely trimmed following laboratory standard procedures and decalcified on a mixture of ≥10–<20% hydrochloric acid (HCl) and formaldehyde (≥3%–<5%) (Osteomoll^®^ rapid decalcifier solution for histology; catalogue no. 101736) over a period of 24–72 h, periodically assessing tissue until it was ready to be further processed. Afterward, longitudinal and sagittal sections were embedded in paraffin wax and cut at 4–5 µm thickness to then be stained with Hematoxylin–Eosin (HE). All slides were reviewed, selecting the most representative section. This was based on a good histological quality and clear invasive front with surrounding nonaffected stroma to evaluate the neoplastic–nonneoplastic transition. The most representative slide was scanned and analyzed through specialized image analysis software (NIS-elements software (Nikon, Tokyo, Japan); Aperio ImageScope (Leica, Wetzlar, Germany)).

### 2.2. Histological Grading

Two grading systems from human (Jesinghaus et al., (2018) [7] and Boxberg et al., (2019) [9]) and veterinary (Nagamine et al., (2017) [2]) medicine were adapted for the present study. The samples were assessed by a blinded diplomat of the American College of Veterinary Pathologists (ACVP) (AC), challenging the adapted systems. 

#### 2.2.1. Invasive front Grading System (IFGS)

An adaptation of Nagamine et al. (2017) grading system for canine oral SCC (OSCC) was used [2], following criteria of degree of keratinization, pattern of invasion, host response, nuclear pleomorphism and mitoses per high power field (HPF), as summarized in Table 2. All features were assessed, focusing exclusively on the invasive front. 

Histological gradings are illustrated in Figure 1, Figure 2, Figure 3, Figure 4 and Figure 5. Degree of keratinization (Figure 1a,b) was assessed from highly keratinized (>50% keratinization, 1 point) to no keratinization (0–5% keratinization, 4 points). Pattern of invasion (Figure 2a,b) ranged from well-differentiated, pushing and infiltrating borders (1 point) up to widespread dissociation into small groups, less than 15 cells (4 points). Host response (Figure 3a,b) was evaluated from a marked inflammatory reaction (1 point), to no inflammation (4 points). Nuclear pleomorphism (Figure 4a,b) was assessed, ranging from little pleomorphism with less than 25% anaplastic cells (1 point) up to extreme nuclear pleomorphism with poor differentiation (75–100% anaplastic cells) (4 points). Mitosis per high power field (HPF) (40×) (Figure 5a,b) ranged from minimal mitotic activity (0–1) (1 point) up to more than 5 mitoses (4 points). Mitosis per HPF was assessed in an overall area of 0.237 mm^2^. Therefore, when assessing 10 HPF (400×), an overall area of 2.37 mm^2^ was covered to guarantee standardization [21]. This procedure was decided for this study as challenged gradings often did not clarify the area covered.

The final addition of the score values of these five morphologic features resulted in a total invasive front score value. Subsequently, the total score value was summarized into four final grades according to Nagamine et al. (2017) [2]: Total score value 6–10 = grade I (well differentiated); Total score value 11–15 = grade II (moderately differentiated);Total score value 16–20 = grade III (poorly differentiated).

#### 2.2.2. Tumor Cell Budding Grading System (TCBGS)

An adaptation from two similar human SCC gradings systems of Jesinghaus et al. (2018) [7] and Boxberg et al. (2019) [9], designed for the uterine cervix and larynx/hypopharynx, respectively, was used. Both systems focus on the invasive front. The features evaluated within this adapted grading included tumor budding in 10 HPF (unspecified covered area), smallest nest size within the invasive front and stromal response associated with the neoplasm (see Figure 6).

The smallest nest size was represented by the complex with least cells within that invasive front (arrows), even if it was only one. In this case, there were single neoplastic cells dissociated from main neoplasm (Giant Schnauzer, No. 33).

In this study, the tumor budding was defined as neoplastic aggregates/complexes of less than five cells that dissociate from the main neoplasm and invade the surrounding stroma. These “complexes” were counted in 10 HPF (40×) in the areas of higher incidence, covering an overall area of 2.37 mm^2^. A numerical value between 1 (no tumor budding) and 3 (≥15 budding foci) was then assigned. Tumor nests, in contrast, included both these smaller (<5 cells) complexes as well as larger aggregates (up to >15 cells) dissociating from the main neoplasm, invading the surrounding stroma. When assessing smallest tumor nest size, a range between more than 15 cells (1 point) and single cell invasion (4 points) was noted (Table 3). 

In this grading system, the two scores were added to the total score value. This total score value divided the neoplasms into well differentiated/grade 1 (total score value: 2–3), moderately differentiated/grade 2 (total score value: 4–5) and poorly differentiated/grade 3 (total score value: 6–7) DCSSC. Additionally, stromal reaction was evaluated, although it was not included into the total score value or final grade. 

### 2.3. Statistical Analysis

Statistical significance analyses were evaluated using IBM SPSS Statistics (version 25). Comparisons between the four genetically based groups (1a–c and 2) were performed with the Kruskal–Wallis test, while the statistical significance between the black and white dogs were analyzed using the Mann–Whitney U test. In the case of the Kruskal–Wallis test, the p-values were adjusted according to Bonferroni. *p* < 0.05 was considered statistically significant.

## 3. Results

Age and sex distribution across phenotypic groups (1a, 1b, 1c and 2) were not significantly different. Similarly, there was no statistical difference between digital tumor localization between limbs, toes and phenotypic group or the presence/absence of neoplastic bone invasion. Nevertheless, the forelimb was the main affected limb, representing 76% of the white-haired dogs and 79% of the dark-haired dogs, where localization was available. 

### 3.1. Invasive front Grading System (IFGS)

Grade I CDSCCs were characterized by well-differentiated, solid neoplastic cords that pushed and infiltrated the surrounding stroma, with abundant keratinization, little anaplasia and barely any mitotic activity, but marked associated inflammation. 

Grade II and grade III, on the other hand, had increasingly poorer differentiation, with multiple small buds that detached from the main neoplasm, with barely to no keratinization, moderate to marked anaplasia and increased mitotic activity but little to no associated inflammation from the host.

According to the IF grading system, 45% of DSSC of the dark-haired animals (Group 1) were grade I, 37.6% were grade II and 20.8% were grade III. In comparison, over three-quarters of light-haired dogs (Group 2) were grade I (77.7%), while the remaining were divided into grade II (16.6%) and grade III (5.5%) (Figure 7A). 

Statistical analysis confirmed that the DSSCs of light-haired dogs (Group 2) were better differentiated, with lower overall grading than the dark-haired dogs (Group 1) (*p* < 0.01, Figure 7A). Therefore, the dark-haired dogs showed significantly less keratinization (*p* < 0.05, Figure 7B), more invasive patterns (*p* < 0.001, Figure 7C) and more marked nuclear pleomorphism (*p* < 0.001, Figure 7D) than their light-haired counterparts. 

When comparing the groups by their genetic haircoat color (see Figure 8), in all statistically significant features (invasive front grading, keratinization, pattern of invasion and nuclear pleomorphism), it was remarkable that the light-haired animals had less malignant characteristics than at least one dark-haired subgroup. Final invasive front grading between light-haired breeds (Group 2) and black & tan dogs (Group 1c), was significantly lower (*p* < 0.05) (Figure 8A). Light-haired dogs had significant more keratinization than black dogs (Group 1a, *p* < 0.05) (Figure 8B). When assessing patterns of invasion, light-haired dogs showed more solid patterns of invasion within the invasive front than black dogs (Group 1a, *p* < 0.05), Schnauzers (Group 1b, *p* < 0.01) and tan and black dogs (Group 1c, *p* < 0.05) (Figure 8C). As far as nuclear pleomorphism within the cells forming the invasive front, the light-haired dogs had a less anaplastic population, with more mature cells than the black dogs (Group 1a, *p* < 0.05) and the Schnauzers (Group 1b, *p* < 0.05) (Figure 8D). However, host response and mitotic activity between groups were not significantly different. 

### 3.2. Tumor Cell Budding Grading System (TCBGS)

Grade 1 CDSCCs in this system were well delineated, with more or less keratinization, either forming solid cords or large groups which detached from the main neoplasm and infiltrated through the invasive front, while exhibiting little associated tumor stroma (scirrhous reaction). On the other hand, grade 2 and 3 CDSCCs were less cohesive, forming small neoplastic aggregates or even individual cells detaching from the main neoplasm and invading the surrounding tissue, which presented a moderate to marked tumor stroma. 

Within the TCB grading system, out of the 76 phenotypical dark-haired animals (Group 1); 45.5% were grade 1; 10.3% were grade 2 and 44.2% were grade 3. Out of the 18 dogs with phenotypical light haircoat (Group 2), the vast majority (88.9%) were grade 1 and 11.1% were grade 3 (Figure 9A). 

There were statistical differences between Groups 1 and 2 compared to the TCB total score (*p* = 0.001) and final grade (*p* < 0.01) (Figure 9). The light-haired dogs (Group 2) had significantly less tumor cell budding in 10 HPF (*p* < 0.001) than dark dogs (Group 1), with fewer buds detaching from the main neoplasia (Figure 9B). When comparing the smallest tumor nest, light-haired dogs had significantly larger nests than their dark counterparts (*p* < 0.001) (Figure 9C). Finally, light-haired animals had significantly less amount of stroma than dark dogs (*p* < 0.001) (Figure 9D).

When comparing each individual morphological feature with its phenotypic haircoat color, depending on the feature evaluated (Figure 10), there were statistical differences between the light-haired dogs (Group 2) and each dark-haired subgroup (1a-1c). The final grade of light-haired dogs, for instance, was statistically lower than the black (Group 1a, *p* < 0.05) and black & tan breeds (1c, *p* < 0.05) (Figure 10A). The number of tumor cell budding foci in Group 2 was significantly lower than Groups 1a (*p* < 0.01), 1b (*p* < 0.05) and 1c (*p* < 0.05) (Figure 10B). When looking at the size of the nests within the invasive front, the light-haired dogs had larger nest sizes than the black (1a, *p* < 0.05), Schnauzers (1b, *p* < 0.05) and black & tan dogs (1c, *p* < 0.01) (Figure 10C). Interestingly enough, even though the amount of tumor stroma was not a part of the numerical score for this system, it was significantly finer in light-haired animals (Group 2), when compared to the Schnauzers (1b, *p* < 0.01) or to the black & tan breeds (1c, *p* < 0.05) (Figure 10D). There was no statistical difference between dark-haired Groups 1a–c in any of the evaluated features. 

### 3.3. Comparison of Invasive Front Grading System and Tumor Cell Budding System 

Both grading systems available were evaluated and compared to each individual factor to evaluate a significant difference between light- and dark-haired canine breeds. There was a significant statistical difference between both phenotypical groups (dark- and light-haired dogs) in these two systems at the final IF grading (*p* = 0.001) and their total score (*p* < 0.01), as well as the degree of keratinization within the invasive front (*p* < 0.05), invasion (*p* = 0.001) and nuclear pleomorphism (*p* < 0.01). Likewise, the different criteria of the TCB system were also statistically significant between the two populations, including tumor budding (*p* = 0.0001), smallest tumor nest (*p* < 0.001) and tumor stroma (*p* = 0.0001), as well as the total cellular dissociation score (*p* = 0.0001) and final grade (*p* = 0.001).

An interesting point was the number of animals having a similar grading on both the IFGS and TCBGS. Groups classified as “well differentiated” (grade I/1) by both systems, was composed of 14 light-haired and 28 dark-haired dogs (Table 4).

When comparing light-haired animals classified as “moderately differentiated” (grade II/2) by both systems, there was no overlap. Interestingly enough, within the dark-haired animals, there were only four dogs graded as “moderately differentiated” by both systems. Digital squamous cell carcinomas from 13 dark-haired animals and one Golden Retriever (case No. 92) were graded as “poorly differentiated” (grade III/3) by both systems.

The CDSCC from three dark-haired dogs (case No. 5, 24 and 25) were graded as “well differentiated” by IFGS, but as “poorly differentiated” by the TCB System. Furthermore, there were CDSCCs from two dogs (case No. 74, a Rottweiler and No. 92, a WHWT) which were graded as “poorly differentiated” by IFGS, while being considered as “well differentiated” by the TCB System. 

## 4. Discussion

There is currently no widely accepted grading system for canine SCC, although one of the most widely used, Broder’s grading system [13], with a 1–4 grade based on the differentiation features, is used to morphologically characterize this tumor [13]. Nonetheless, given that there is no prognostic significance and often somewhat subjective assessment, many pathologists fail to characterize it. Broder’s system was not included in this case since, similar to what other studies showed [11], there can be different grades of differentiation within the same tumor, making it hard when evaluating the sample. In this study, the goal was to compare two different adapted grading schemes to ascertain that there was a morphological disparity between light- and dark-colored animals with CDSCC and some kind of grading congruence between both systems. This allows a better comprehension of CDSCC and, hopefully, the development of a future grading system with prognostic correlation. 

Canine digital cell carcinomas (CDSCC) are known to be particularly aggressive when compared to other cutaneous locations. Even though these neoplasms are more frequently seen in classically dark-haired breeds, there is no literature available examining if these tumors are morphologically different than their light-haired counterparts, suggesting different, maybe more aggressive, behavior. Through the adaptation of both human (Jesinghaus et al. (2018) [7] and Boxberg et al. (2019) [9]) and veterinary (Nagamine et al., (2017) [2]) SCC grading systems, we evaluated CDSCC from animals of both haircoat colors and investigated if there was any statistical difference between the different morphological features based on their presumed genetic and phenotypical haircoat color. 

Skin pigmentation is a point of interest in human medicine regarding evaluating susceptibility to certain skin neoplasias, such as cutaneous melanoma, or basal cell carcinoma [22]. Additionally, some skin tumors, such as melanoma and nonmelanoma skin cancers, are more often seen in white populations, believed to be closely associated with skin color and UV-light exposure (among other factors) [23]. This is also postulated because skin cancers are less common in People of Color than in Caucasians [23,24,25]. Nonetheless, this has not been widely studied in veterinary medicine. In canine melanomas, postulated to be a potential human model [26], several copy number alterations and low numbers of single-nucleotide variations with non-UV-associated mutations were identified [26]. In both dogs and humans, mitogen-activated protein kinase (MAPK) and phosphoinositide 3-kinase (PI3K) were associated with mucosal melanoma [26]. Nevertheless, there are currently no detailed studies about comparing canine squamous cell carcinoma (digital and nondigital) with haircoat color and taking into account the speculated genotypic haircoat. 

Breeds represented in our study were mainly dark breeds, the most common being Schnauzers and Rottweilers, and a markedly smaller light-haired population, similar to the literature [11]. When comparing the localization of this tumor in each subgroup, either the limb or toe, there were no statistical differences between groups (*p* < 0.05). Unfortunately, there was a great number of animals with no information regarding location, therefore, the interpretation of end results in this parameter must be taken with caution. 

To follow the International Tumor Budding Consensus Conference (ITBCC), 2016, of colorectal cancer [27], and other similar studies [9] with the objective to increase reproducibility on a diagnostic setting, slides were assessed on HE alone. This approach was taken because meta-analyses suggest that the prognostic evidence assessed on HE vs. immunohistochemistry (IHC) is not significant, although IHC may allow a higher interobserver agreement [27].

When looking at the light-haired dogs CDSCC, these were more frequently well differentiated, with abundant keratinization, well-formed, pushing solid cords infiltrating within the invasive front, little anaplasia and rare, if any, mitotic activity. On the other hand, the darker breeds often had more “malignant” features, with frequent budding, less keratinization and more anaplasia within the invasive front. This is particularly interesting for Group 1b, the Schnauzers, which were the most homogenous with all the same breed of animals, leading to speculation that, when encountering a more poorly differentiated CDSCC, it is more likely to be from a dark-haired animal, although the underlying reason for this is yet to be elucidated. 

When comparing the two systems provided, within the dark-haired population, IFGS showed 41.5% grade I, 37.6% grade II and 20.8% grade III. On the other hand, TCBGS in the same population showed a proportion of 45.5% grade I 10.3% grade II and 44.2% grade III. This illustrates that, for well-differentiated CDSCC, with a solid pattern of invasion, well-keratinization and low mitotic activity tends to be engulfed as a low grade/grade I by both systems. Nevertheless, when more malignant features are present, the IF grading system tends to include it as grade II, while the TCB grading system would more likely assign it to be grade III. Interestingly enough, out of all the animals, there were only three cases that were graded as “well differentiated” by the IF system, while having a “poorly differentiated” grade on the TCB System. On the other hand, there were two cases characterized as “poorly differentiated” by the IF system, while being graded as “well differentiated” by the TCB grading system. This apparent incongruence could be explained by different features evaluated within the invasive front, which rarely overlap in both systems. Also, it can be explained by the marked importance that the TCB grading system gives to nest size and budding (2/3 features evaluated), while the IF grading system only pays attention to this feature in one out of the five characteristics evaluated, hence the grading disparity in some cases. It must be pointed out that, although the single cell tumor nests are of great importance, the less cohesive these cells are, the more poorly differentiated the neoplasm is likely to be; therefore, a more aggressive behavior can be hypothesized. 

Furthermore, the IF grading system pays special attention to additional features, which are also theoretically associated with the pathogenesis of this neoplasm, such as host response. An inflammatory reaction secondary to a tumor invasion, particularly in those tumors that produce extracellular keratin (and, therefore, presumably better differentiated), is expected to elicit a profound immune response. On the other hand, neoplastic tactics of immune-tolerance mechanisms and immune-response evasion were shown to modulate the inflammatory response by attenuating it and allowing tumors to create a favorable microenvironment for invasion [27,28,29]. Nonetheless, similar to another small study [28], our results concluded that there was no significant difference between degree of host response/inflammation between the groups. Additionally, it has to be pointed out that the IFGS, even when used in CDSCC, is an adaptation of Nagamine et al.’s system [2] which, in itself, is an adaptation from human medicine [30] and, henceforth, certain features cannot be extrapolated. For instance, dogs with masses on toes (either inflammatory or neoplastic), will tend to inflict self-trauma, either through chewing or licking, thus causing a secondary inflammatory response. This would also explain why inflammation among groups may not be significant, given that all animals may traumatize the area one way or another. 

Additionally, other interesting features, such as stromal reaction, were evaluated within the TCBGS, although did not play a role in the total score or final grading. This fibrovascular scaffold, which includes fibroblasts, vasculature, extracellular matrix and other extracellular molecules, set the tumor-microenvironment that favors tumor growth and expansion through different mechanisms [31]. Taking into account this particular morphological feature within the grading may be a representation of the tumor microenvironment, thus becoming more prominent in those less differentiated with, hypothetically, more aggressive behavior. 

When performing the gradings, there were statistical differences between the light- and the dark-haired breeds (which were represented by phenotypically black & tan breeds, black breeds and black Schnauzers) in both IF score and grading (*p* < 0.01 and *p* < 0.01, respectively) and TCB score and grading (*p* < 0.01 and *p* < 0.01, respectively) systems. Additionally, when comparing each individual morphologic feature, there were statistical differences in degree of keratinization within the invasive front, pattern of invasion, nuclear pleomorphism, tumor budding activity in 10 HPF, smallest nest size and amount of tumor stroma. These features were consistently better differentiated in light-haired rather than dark-haired breeds. Interestingly, there was not an overall significant difference between the phenotypical dark-haired groups (presumed genotypes KB/KB, KB/Ky, ky/ky and at/*) in any of the scoring systems or individual features. This finding highly suggests that dark-haired breeds tend to have more morphologically poorly differentiated CDSCC when compared to light-haired breeds, although different biological behavior cannot be predicted, only hypothesized. It would be interesting to know the prognosis of these tumors based on the histomorphological features of the invasive front. Sadly, no follow-up information was available concerning the samples evaluated in our study.

As mentioned before, the colors of the haircoat and claws depend on the content/absence and distribution of eumelanin in these structures. This pigment distribution and content depend on simple genetic variants of the E-, K- and A-locus (among others) [18]. In our study, all dogs had concordant color in both haircoat and claws, with black claws in dark-haired animals (Group 1) and light claws in light-haired animals (Group 2). Altogether, it could be inferred that the poor differentiation of CDSCC, which were associated with the haircoat color (most obvious in dark-haired animals), is also similarly associated with the claw pigmentation. In summary, this “poor differentiation” of CDSCC could potentially be associated with the eumelanin biochemistry of processing and incorporation of this pigment into the claw. 

In general, when assessing adapted SCC grading systems, a few limitations have to be taken into account. To begin with, the best/most malignant invasive front in each CDSCC did not always match the deepest invasive front (as evaluated in other publications [2,6,7,9]), as sometimes this was located within the bone, while others were within the dermis. This allows us to speculate that, due to different cellularity and structure of the surrounding normal stroma (either bone or dermis), the neoplastic cells may render different strategies of invasion, making it inconsistent during the grading. Also, since the deep invasive front was not always the most malignant front, this may theoretically support the tumor biological behavior and spread, infiltrating in all directions.

A further limitation encountered was the different grades of differentiation depending on the area within the tumor, as reported in the literature [11]. Also, due to the morphological overlap between certain digital squamous cell carcinomas and less malignant epithelial neoplasms, such as subungual keratoacanthoma, some well-differentiated squamous cell carcinomas within this location may be overlooked or misdiagnosed, thus underestimating the prognosis. In this particular study, the light-haired population (*N* = 18) was much smaller than the phenotypical dark-haired one (*N* = 76), which makes interpretation between phenotypical groups somewhat difficult. Nevertheless, this disparity is concordant with previous literature [11], where dark dogs are prone to this neoplasm and, consequently, are more numerous. 

Additionally, when assessing tumor cell nesting, complete cellular dissociation of the tumor aggregates from the main neoplasia has to be assumed. This can be somewhat problematic given that this is a 2D assessment (a histological slide) of a 3D event, never making sure that the small complexes might be connected to the primary mass in deeper sections or when a different orientation is given. This dissociation, however, has to be assumed when assessing the invasive front, given that there is currently no other available system.

This study opens up interesting future research concerning CDSCC, such as the different prognosis based on these neoplasms’ histomorphological features. Currently, there is no available grading system for CDSCC that provides prognostic clinical insight. Additionally, the question regarding whether phenotypically dark breeds, with their presumed genotypical haircoat color, are genetically predisposed to a more morphologically poorly differentiated CDSCC or whether light-haired breeds have genetic protection against this tumor still remains to be studied through future genetic analysis. Throughout this study, no true genetic haircoat analysis was made. The assumption of the presumed genotype was only based on the phenotypical color. Nevertheless, this may result in a scaffold for future research studies in which true hair-coat genetic analysis can be performed. 

## 5. Conclusions

To our knowledge, this is the first study comparing CDSCC in dogs by two histological grading systems, taking into account their phenotypical and presumed genotypical haircoat color and demonstrating that digital squamous carcinomas are not only more common in dark-haired dogs, but potentially more aggressive. When comparing both challenged TCB and IF grading systems, they often overlapped when grading well-differentiated tumors. On the other hand, when more “malignant” features were present in the CDSCC, the classification systems often placed them in different grade (II vs. III). With this study, conclusions regarding the most accurate grading system for CDSCC cannot be drawn, since no outcome was available in any of the cases. 

## Figures and Tables

**Figure 1 vetsci-08-00003-f001:**
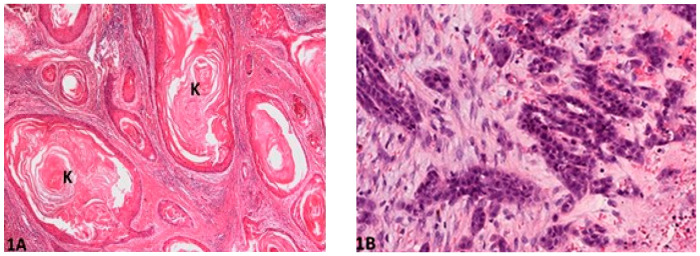
Degree of keratinization: (**A**) (O.M. 2×): Well-differentiated and highly keratinized cells (K) with over 50% of keratinization (Gordon Setter, No. 49); (**B**) (O.M. 10×): Less than 5% cells exhibiting keratinization (Gordon Setter, No. 55). O.M: original magnification.

**Figure 2 vetsci-08-00003-f002:**
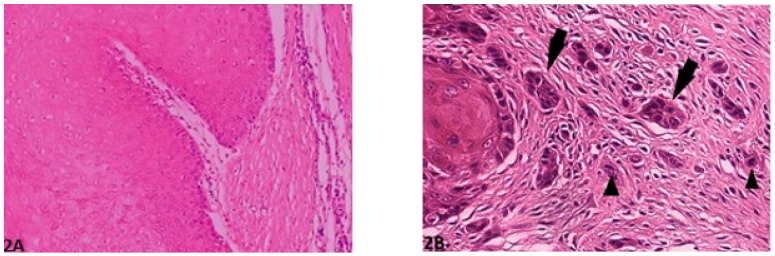
Pattern of invasion: (**A**) (O.M. 8×): Well-differentiated, expansively growing tumor borders compressing the surrounding stroma (Golden Retriever, No. 87); (**B**) (O.M. 20×): Wide-spread cellular dissociation in small groups (<15 cells, arrow) and/or single cells (arrow heads, medium Schnauzer, No. 45). O.M: original magnification.

**Figure 3 vetsci-08-00003-f003:**
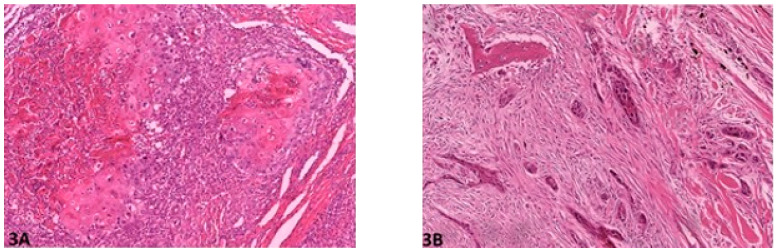
Host response: (**A**) (O.M. 8×): Marked inflammation, occasionally obliterating the neoplastic cells (Russian Terrier, No. 6); (**B**) (O.M. 8×): Minimal to virtually no associated inflammation (Rottweiler, No. 70). O.M: original magnification.

**Figure 4 vetsci-08-00003-f004:**
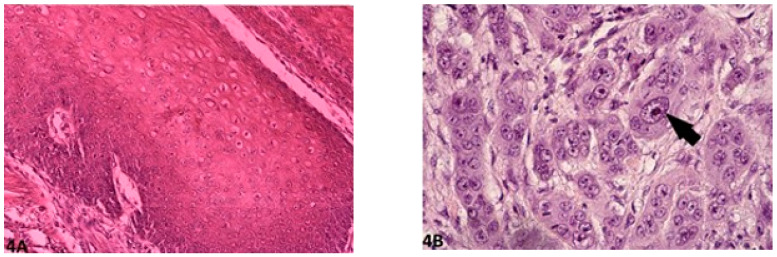
Nuclear pleomorphism: (**A**) (O.M. 8×): Mild/minimal pleomorphism in <25% neoplastic cells (Golden Retriever, No. 87); (**B**) (O.M. 40×): Extreme nuclear pleomorphism with intense nuclear atypia (arrow), accounting for more than 75% of neoplastic cells (Giant Schnauzer, No. 56). O.M: original magnification.

**Figure 5 vetsci-08-00003-f005:**
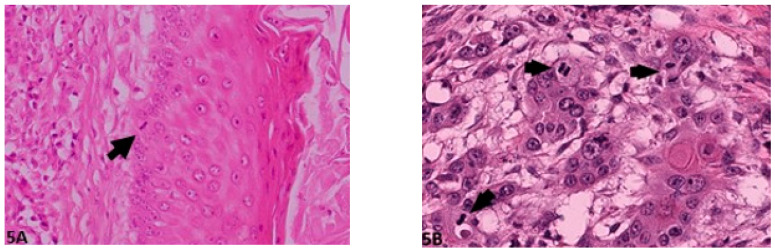
Mitosis per HPF: (**A**) (O.M. 40×): Single mitotic figure (arrow, Gordon Setter, No. 50); (**B**) (O.M. 40 x): Multiple mitotic figures (arrows, Gordon Setter, No. 56). O.M: original magnification.

**Figure 6 vetsci-08-00003-f006:**
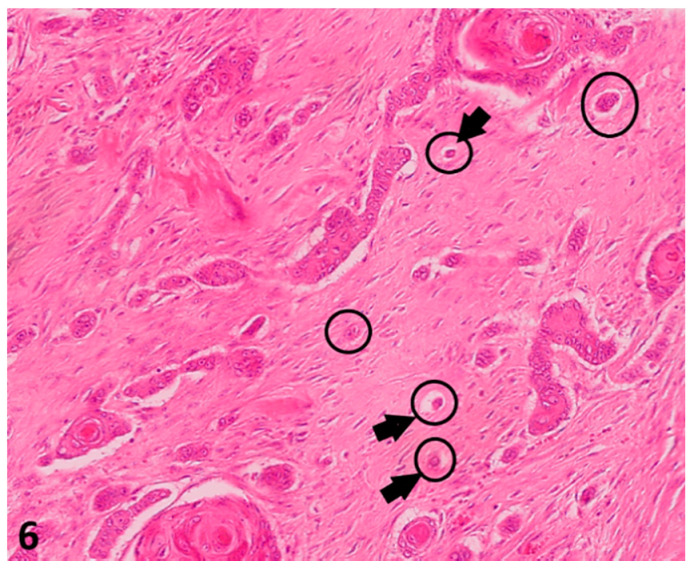
(O.M. 4×): Histological pictures (Hematoxylin–Eosin (HE) stain) illustrating tumor budding in a canine digital squamous cell carcinoma: Only complexes of less than five cells were counted in 10 HPF (40×) in the area of biggest incidence (delineation) within the invasive front. O.M: original magnification.

**Figure 7 vetsci-08-00003-f007:**
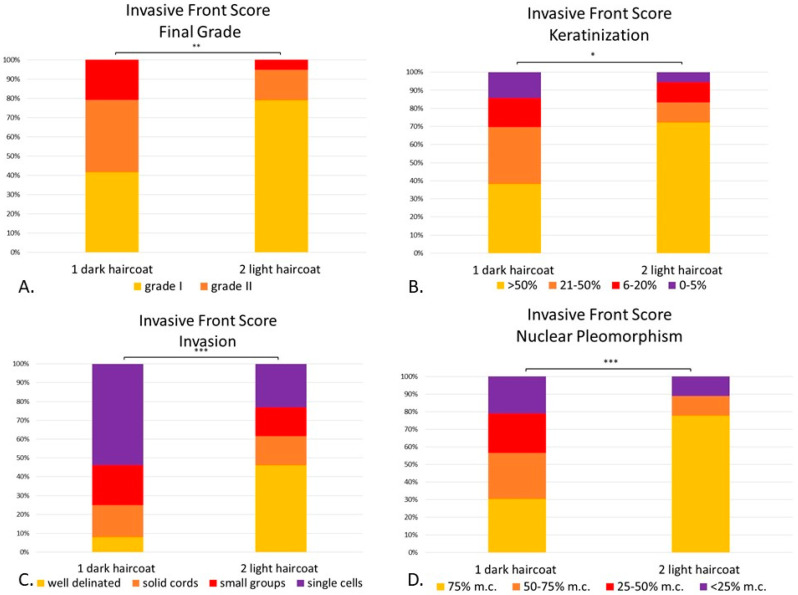
(**A**–**D**). Canine digital squamous cell carcinoma. Statistical differences between phenotypically dark-haired (Group 1) and light-haired (Group 2) dogs according to the invasive front system. There were statistical differences between Group 1 and 2 in final invasive front grading (*p* < 0.01) (**A**), amount of keratinization within the invasive front (*p* < 0.05) (**B**), pattern of invasion (*p* < 0.001) (**C**) and degree of nuclear pleomorphism (*p* < 0.001) (**D**). Mitoses and inflammation were not included given that they were not statistically significant between groups. * *p* < 0.05; ** *p* < 0.01, *** *p* < 0.001. Abbreviations: m.c.: mature cells.

**Figure 8 vetsci-08-00003-f008:**
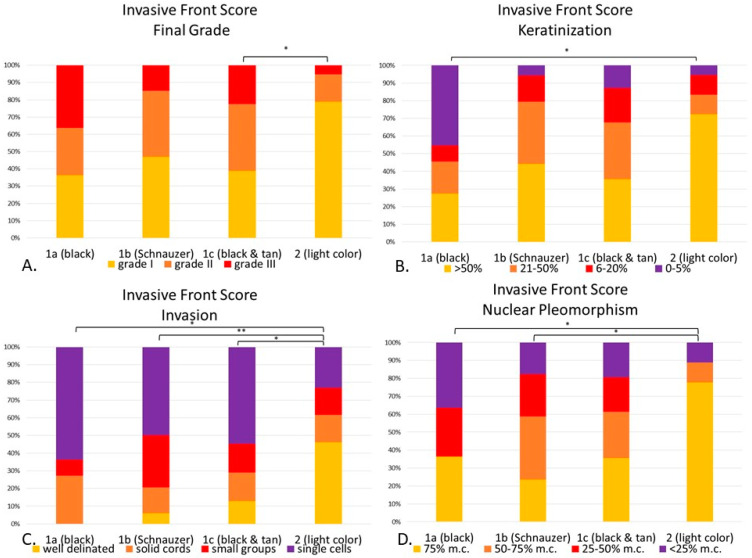
(**A**–**D**). Canine digital squamous cell carcinoma. Statistical differences between genetically defined subgroups (1a–c) and Group 2 according to the invasive front grading system. Invasive front final grade (*p* < 0.05), with statistical differences between Group 1c and 2 (**A**). Keratinization (*p* < 0.05) with differences between 1a and 2 (*p* < 0.05) (**B**). Invasion (*p* < 0.01) with significant differences between 1a (*p* < 0.05), 1b (*p* < 0.01) and 1c (*p* < 0.05) when compared to 2. (**C**). Nuclear pleomorphism (*p* < 0.05) with significant differences between 1a (*p* < 0.05) and 1c (*p* < 0.05) when compared with Group 2. (**D**). * *p* < 0.05; ** *p* < 0.01. Abbreviations: m.c.: mature cells.

**Figure 9 vetsci-08-00003-f009:**
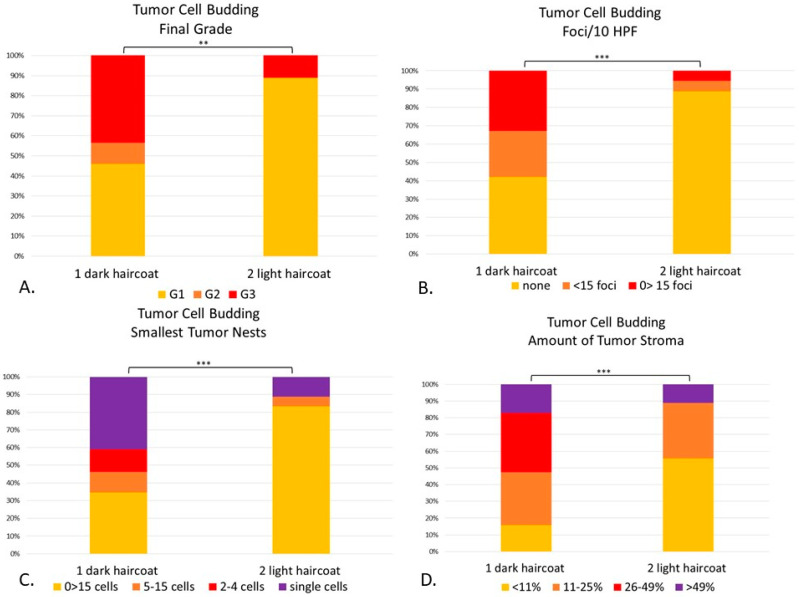
(**A**–**D**). Canine digital squamous cell carcinoma. Statistical differences between phenotypically dark-haired (Group 1) and light-haired (Group 2) dogs according to the tumor cell budding system. There were statistical differences between Groups 1 and 2 in final grade (*p* < 0.01) (**A**), tumor cell budding in 10 HPF (*p* < 0.001) (**B**), smallest tumor nest size (*p* < 0.001) (**C**) and amount of tumor stroma (*p* < 0.001) (**D**). ** *p* < 0.01; *** *p* < 0.001.

**Figure 10 vetsci-08-00003-f010:**
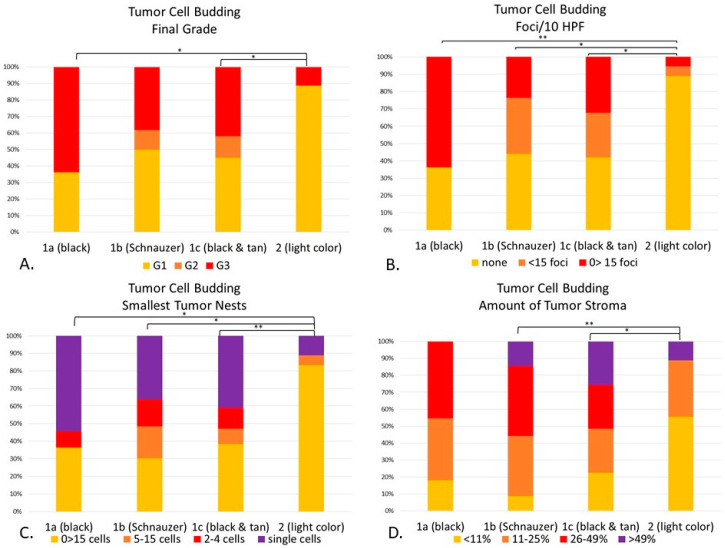
(**A**–**D**). Statistical differences between genetically defined subgroups (1a–1c) and Group 2 according to the tumor cell budding system. Cellular dissociation final grade (*p* < 0.01), with statistical differences in Group 1a (*p* < 0.05), and 1c (*p* < 0.05) when compared to Group 2 (**A**). Budding/10 HPF (*p* < 0.01) was different between Groups 1a (*p* < 0.01), 1b (*p* < 0.05) and 1c (*p* < 0.05) when compared to Group 2 (**B**). Smallest tumor nest size (*p* < 0.01) was significantly different between Groups 1a (*p* < 0.05), 1b (*p* < 0.05) and 1c (*p* < 0.01) when compared to Group 2 (**C**). The amount of tumor stroma (*p* < 0.01) was statistically different between Groups 1b (*p* < 0.01) and 1c (*p* < 0.05) when compared to Group 2. There were no statistical differences among dark breeds when compared to each other. (**D**) The amount of tumor stroma of Group 2 was statistically significant when compared to Group 1b (*p* < 0.01) and 1c (*p* < 0.05). * *p* < 0.05; ** *p* < 0.01.

**Table 1 vetsci-08-00003-t001:** Dog signalment and affected region with digital squamous cell carcinoma (DSCC) in the present study.

Group Assignment	Phenotypic Haircoat Color	Breed	No. of Dogs	Mean Age (y.o.)	Sex					Number of Dogs with Affected Limb RF/RH/LF/LH/U
					M	MC	F	FS	U	
**Group 1a** (*n* = 11) No. 1–11	Black	Russian Terrier	7	9.5	2	0	1	2	2	1/0/3/0/3
		Briard	2		0	1	1	0	0	0/0/2/0/0
		Giant Poodle	1		0	1	0	0	0	0/0/1/0/0
		Labrador Retriever	1		0	1	0	0	0	0/0/1/0/0
**Group 1b** (*n* = 34) No. 12–45		Giant Schnauzer	27	8.5	10	8	4	5	0	9/1/10/2/5
		Standard Schnauzer	7		3	1	1	2	0	0/3/2/2/0
**Group 1c** (*n* = 31) No. 46–76	black & tan	Rottweiler	21	9.6	5	2	5	7	2	7/1/0/2/6
		Gordon Setter	10		6	1	2	1	0	3/2/0/0/5
**Group 2** (*n* = 18) No. 77–94	Light	Golden Retriever	15	10.5	8	1	3	0	3	4/1/4/2/4
		WHWT	3		2	0	1	0	0	0/0/2/0/1

Abbreviations: y.o.: years old; M: male intact; MC: male castrated; F: female intact; FS: female spayed; U: unknown; RF: right forelimb; RH: right hindlimb; LF: left forelimb; LH: left hindlimb; U: unknown; WHWT: West Highland White Terrier.

**Table 2 vetsci-08-00003-t002:** Invasive front grading system (IFGS) used in the present study of canine digital squamous cell carcinoma (CDSCC) (adapted from Nagamine et al., (2017) [2] for use in canine oral squamous cell carcinoma).

Morphological Feature	Score Value			
	1	2	3	4
Degree of keratinization	Highly keratinized (>50% cells)	Moderately keratinized (20–50% of cells)	Minimal keratinization (5–20% of cells)	No keratinization (0–5% of cells)
Pattern of invasion (bone or dermis)	Pushing, well-differentiated, infiltrating borders	Infiltrating, solid cords, bands and/or strands	Small groups/cords of infiltrating cells (*n* > 15)	Widespread cellular dissociation in small groups and/or in single cells (*n* < 15)
Host response	Marked	Moderate	Slight	None
Nuclear pleomorphism	Mild (<25% anaplasia)	Moderate (25–50% anaplasia)	Marked (50–75% anaplasia)	Extreme (75–100% anaplasia)
Mitosis HPF (40×)	0–1	2–3	4–5	>5

**Table 3 vetsci-08-00003-t003:** Tumor cell budding system used in our study to determine tumor grade based on tumor budding activity and cell next size score adapted from human cervical squamous cell carcinoma (SCC) (Jesinghaus et al., 2018) [7] and laryngeal/hypopharyngeal SCC (Boxberg et al., 2019) [8].

Tumor Budding Activity/10 HPF	Score Value
No budding	1
<15 budding foci	2
≥15 budding foci	3
**Smallest cell nest size**	
>15 cells	1
5–15 cells	2
2–4 cells	3
Single cell invasion	4

**Table 4 vetsci-08-00003-t004:** Number of animals with digital squamous cell carcinoma in both invasive front grading system and tumor budding system.

	Invasive Front Grading System			Tumor Cell Budding Grading System			Number of Dogs with Same Grading by Both Systems	
	Light-Haired *n* = 18	Dark-Haired *n* = 76		Light-Haired *n* = 18	Dark-Haired *n* = 76		Light-Haired *n* = 15	Dark-Haired *n* = 45
Grade I *n* = 47	15	32	Grade 1 *n* = 51	16	35	Well differentiated *n* = 42	14	28
Grade II *n* = 32	3	29	Grade 2 *n* = 8	0	8	Moderately differentiated *n* = 4	0	4
Grade III *n* = 17	1	16	Grade 3 *n* = 35	2	33	Poorly differentiated *n* = 14	1	13

## Data Availability

Data available on request due to restrictions eg privacy or ethical.

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
