# Peer review of "Evaluating the Histologic Grade of Digital Squamous Cell Carcinomas in Dogs with Dark and Light Haircoat—A Comparative Study of the Invasive Front and Tumor Cell Budding Systems"

_vetsci, 2020, doi:10.3390/vetsci8010003_

Round 1
Reviewer 1 Report
The manuscript “Invasive Front Grading and Tumor Cell Budding in Digital Squamous Cell Carcinoma in Dogs with Dark and Light-haired Color” gives interesting information for clinicians and pathologists, however it needs to be improved before its publication in this journal.
Main concerns:
The discussion needs to be revised for an easier lecture and comprehension. The beginning needs to explain that for the authors think that this study demonstrates that there is not a perfect classification for CDSCC. However, the study clarify the clinical and pathological situation of this kind of tumor in dogs, so later the authors need to explain the additional information that give this manuscript compared with previous information.
In all the manuscript (text, tables and figures) it’s necessary to clarify about which tumor you are talking at that moment. So, to be sure about abbreviations that were used: CDSCC, SCC, DSCC, etc.
Minor concerns:
Line 11: Canine digital SQUAMOUS cell carcinomas (CDSCC) 

Line 35: tumors in veterinary species including papilloma-induced neoplasms and chronic ultraviolet (UV)-damage, 

Lines 40-41 and in all the text, tables and figures: adjective in minuscule and breed in capital as “large breeds such as giant Schnauzer, black Labrador Retrievers and 
standard Poodles..” 

Lines 47-48: needs reference
Lines 48-53 and in all the text, tables and figures: “well differentiated/grade I”, “moderately differentiated/grades II and III”
Line 65 
and in all the text, tables and figures: no abbreviations at the beguining of sentence
Lines 66-68: needs reference
Line 71: “appearance [17].”
Line 76: KITLG….name complete with the abbreviation

Lines 80-83: “The objectives of this study were (1) compare two adapted grading schemes from both human and veterinary medicine for canine digital squamous cell carcinoma; (2) evaluate if there are significant characteristic disparity between light and dark 
coated dogs, based on the grading schemes elaborated and taking into account their genotypical hair coat color.”
Lines 86-89: add the number exact to the percentage like 49% (¿?/2,983), etc.
In all the text, tables and figures: use dog or dogs instead of animal or animals
Lines 97-104: In group 1, the phenotypically “dark-haired breeds”, composed of 76 dogs, was further divided into 3 subgroups: group 1a (n = 11) was made by genetically entirely black breeds (KB/KB) including 7 Russian Terrier, 2 black Briard, 1 black giant Poodle and 1 black Labrador Retriever; group 1b (n= 34) were Schnauzer (KB/KB and KB/KY) represented by 27 giant Schnauzer and 7 black standard Schnauzer; and group 1c (n= 31) consisted of genetically “black and tan” (KY/KY + at/*) breeds, which was represented by 21 Rottweiler and 10 Gordon Setter. Group 2 (n= 18) were the light-colored breeds (e/e) including 15 Golden Retriever and 3 West Highland white Terrier.
Lines 128 and 135, and in all the text, tables and figures: “et al.” in cursive and use “figure 1” (in the text) instead “Fig.1” (only just into brackets)
Line 193 and in all the text, tables and figures: “this study” instead “our study”
Table 3: put “Smallest cell…” on the left side
Line 215: p<0.05
All Figures: the texts under the images are too small
Line 342: “one” instead “on”
Line 473: “… available grading system for DSCC in dogs that would give a prognostic 
…”
Author Response
- Thank you for the rapid assessment and constructive comments concerning the paper. We have addressed all issues and corrected most of them by an addition within the paper. There are some minor comments that we would like to get clarification on, so we can make the changes accordingly. Thank you for your time,
The manuscript “Invasive Front Grading and Tumor Cell Budding in Digital Squamous Cell Carcinoma in Dogs with Dark and Light-haired Color” gives interesting information for clinicians and pathologists, however it needs to be improved before its publication in this journal.
Main concerns:
The discussion needs to be revised for an easier lecture and comprehension. The beginning needs to explain that for the authors think that this study demonstrates that there is not a perfect classification for CDSCC. However, the study clarify the clinical and pathological situation of this kind of tumor in dogs, so later the authors need to explain the additional information that give this manuscript compared with previous information.
- The point of the paper is to compare to different grading schemes, to be able in later studies to see if there is a perfect grading system (once there is clinical information associated with the cases). The intention of the manuscript is to compare two different schemes, as a perfect grading system cannot be assessed with the available data.
- We have included this in the beginning of the discussion to make it more clear: “ In this study, the goal was to compare two different adapted grading schemes to compare if there was a morphological disparity between light and dark-colored animals with DSCC and grading congruence between both systems. This allows to a better comprehension of CDSCC and, hopefully, a development of a future grading system with prognostic correlation. “
In all the manuscript (text, tables and figures) it’s necessary to clarify about which tumor you are talking at that moment. So, to be sure about abbreviations that were used: CDSCC, SCC, DSCC, etc.
- This has been corrected using consistently CDSCC when referring to DSCC on the study animals.
Minor concerns:
Line 11: Canine digital SQUAMOUS cell carcinomas (CDSCC)
- This has been corrected 

Line 35: tumors in veterinary species including papilloma-induced neoplasms and chronic ultraviolet (UV)-damage,
- This has been corrected 

Lines 40-41 and in all the text, tables and figures: adjective in minuscule and breed in capital as “large breeds such as giant Schnauzer, black Labrador Retrievers and 
standard Poodles..”
- This has been corrected 

Lines 47-48: needs reference
- This has been added
Lines 48-53 and in all the text, tables and figures: “well differentiated/grade I”, “moderately differentiated/grades II and III”
- The correspondence of grade I/well-differentiated, II/moderately differentiated and III/differentiated was included in the material and methods. Given that this scheme (lower numbers better differentiated than higher numbers) are well-established in both human and veterinary medicine, we feel that repeating the long nomenclature each time would be redundant, thus making the reading process less “light” for the reader. We have used occasionally used “well/moderately/poorly differentiated” nomenclature instead when trying to highlight a certain morphological feature or when there were many numbers involved (to not overcrowd the text with numbers if there was no need).
Line 65 
and in all the text, tables and figures: no abbreviations at the beginning of sentence
- This has been corrected
Lines 66-68: needs reference
- We are unsure if this is the phrase of concern, as there is a reference attached: “Even though CDSCC is a fairly frequent diagnosis, accounting for the most common neoplasia in the canine digit, with an incidence up to 47.4% of all malignant digital tumors [16] there is not much literature available [11].”
Line 71: “appearance [17].”
- This has been corrected
Line 76: KITLG….name complete with the abbreviation

- This has been corrected
Lines 80-83: “The objectives of this study were (1) compare two adapted grading schemes from both human and veterinary medicine for canine digital squamous cell carcinoma; (2) evaluate if there are significant characteristic disparity between light and dark 
coated dogs, based on the grading schemes elaborated and taking into account their genotypical hair coat color.”
- This has been corrected
Lines 86-89: add the number exact to the percentage like 49% (¿?/2,983), etc.
- This has been corrected
In all the text, tables and figures: use dog or dogs instead of animal or animals
- In tables and images, it has been changed to dogs. For the general text, since dog is a word that appears multiple times, animal has still been used in some cases just to make a lighter read, since it´s understands that animal and dog can be used interchangeably in this paper.
Lines 97-104: In group 1, the phenotypically “dark-haired breeds”, composed of 76 dogs, was further divided into 3 subgroups: group 1a (n = 11) was made by genetically entirely black breeds (KB/KB) including 7 Russian Terrier, 2 black Briard, 1 black giant Poodle and 1 black Labrador Retriever; group 1b (n= 34) were Schnauzer (KB/KB and KB/KY) represented by 27 giant Schnauzer and 7 black standard Schnauzer; and group 1c (n= 31) consisted of genetically “black and tan” (KY/KY + at/*) breeds, which was represented by 21 Rottweiler and 10 Gordon Setter. Group 2 (n= 18) were the light-colored breeds (e/e) including 15 Golden Retriever and 3 West Highland white Terrier.
- The usage of words for numbers between 0-10 and digits for numbers above 11 is based on “how to write a scientific paper” according to cnx.org (https://cnx.org/contents/k23x5aEB@1/Rules-for-the-Use-of-Numbers-in-Scientific-Writing). Nevertheless, if the editor believes this should be change, we are happy to do so.
Lines 128 and 135, and in all the text, tables and figures: “et al.” in cursive and use “figure 1” (in the text) instead “Fig.1” (only just into brackets)
- This has been corrected
Line 193 and in all the text, tables and figures: “this study” instead “our study”
- This has been corrected
Table 3: put “Smallest cell…” on the left side
- This has been corrected
Line 215: p<0.05
- This has been corrected
All Figures: the texts under the images are too small
- This has been corrected
Line 342: “one” instead “on”
- This has been corrected
Line 473: “… available grading system for DSCC in dogs that would give a prognostic 
…”
- This has been corrected
Reviewer 2 Report
Overall, this is a well-done, interesting paper which shows the applicability of novel grading schemes in veterinary medicine. As the authors mentioned themselves, the utility of the grading schemes is limited by lack of follow up and therefore prognostic significance. In addition, there are some major points which should be addressed:
1 – The subgrouping of the dark canine breeds is listed differently in the abstract than in the body of the paper. (1a is black/tan in abstract but black in the body of the paper itself)
2 – The interchangeable use of phenotypic (hair / nail color) and genotypic (ee / EE) is problematic. The authors are relying on clinic notes of the breeds and not a genetic analysis of the dogs themselves. And many times the breeds can be aspirational as opposed to true (in other words not all “pure bred” dogs are indeed pure bred). The analyses should be therefore limited to what the authors know (color) and not what is being presumed (genetics). A discussion of possible genetic relationship to the tumor malignancy is bot appropriate and necessary at the conclusion, but should be limited to speculation.
3 – While the authors mention the Broder’s scheme, some orientation for the reader as to how that scheme works relative to the other schemes is warranted. Additionally, a more clear rational for why this scheme is inadequate should be given (other than the rational in human medicine). Has this scheme been proven not to be prognostic? Were these tumors graded by this scheme on the initial pathology report? How did it compare to the IF and TCB schemes?
4 – For the TCB, how do you rule out that the nests are not 2 dimensional sections through finger like (therefore attached) three dimensional extensions of the tumor? Are you assuming dissociation?
5 – Finally, while the authors describe the few discrepancies between the two schemes, a clear guidance of which is preferred is not given. If the authors believe that a clear preference cannot be made without follow, this should be clearly stated.
Smaller areas:
There are several places where the use of language is misleading / confusing. Some examples:
- The title does not make clear there are 2 grading schemes evaluated. Consider: Grading of DSCC in Dogs with light and dark hair by invasive front and tumor cell budding.
- Line 42: “location, but papilloma virus has not been demonstrated by PCR positivity in affected digits"
None of the photos include size markers or magnification
CDSCC and DSCC are used interchangeably. Please make sure that abbreviations are consistent throughout paper.
Line 54: Since invasive front is a relatively new concept, a brief orientation such as “invasive front (the area of the margin of the neoplasm consisting of the tumor-host interface)” would be useful to the non-pathologist reader
Line 83: “genotypic” as mentioned above you are evaluating the visible, phenotypic, hair color only. I suggest removing the (presumed) genetics of the dogs in Table 1 in addition to making the subtypes consistent with the abstract
Line 134: Nagamine’s paper specifically does state that the invasive front was evaluated (hence the title of the paper and grading scheme)
Lines 146-148: THANK YOU for providing a standardized area which should be (but is not) specified in all histologic grading scheme!
Figures 2-5 – appear twice (although this may be the editors formatting error)
Line 171: “occasionally obliterating the neoplastic”
Figure 4 – use photos of the same magnification
Figure 6 – I think the legend was mis-formatted (again, editor error) and that lines 190-192 should be part of the legend and not text
Lines 197-199: awkward wording. Suggest: “Tumor nests, in contrast, include both these smaller (<5cell) complexes as well as larger aggregates (>15 cells) dissociating . . .” Also, per pint 5 above, consider changing dissociating to “locally separate” or something similar to acknowledge the limitations of 2D histologic sections
Figure 7 – Even in the mitoses and inflammation were not significant, it would be useful to include that information
Line 322 – Wording obscures that these schemes evaluate completely different criteria. “Likewise, the difference criteria of the TCB scheme were also statistically significant between the two populations” (or similar)
Line 339 – what color are dogs 74 and 92? So the biggest difference between the schemes was the separation of grades 2 and 3 with a 5 (5,24,25,74,92) going from high to low? Dark more likely to be well differentiated in TCB and light more likely to me poorly differentiated by TCB?
Lines 360-361 People of Color (capitalize)
Discussion: extremely well done. Would just like to see more of a definite answer on preference or lack thereof
Author Response
Thank you for the rapid assessment and constructive comments concerning the paper. We have addressed all issues and corrected most of them by an addition within the paper. There are some minor comments that we would like to get clarification on, so we can make the changes accordingly.
Thank you for your time.
Overall, this is a well-done, interesting paper which shows the applicability of novel grading schemes in veterinary medicine. As the authors mentioned themselves, the utility of the grading schemes is limited by lack of follow up and therefore prognostic significance. In addition, there are some major points which should be addressed:
1 – The subgrouping of the dark canine breeds is listed differently in the abstract than in the body of the paper. (1a is black/tan in abstract but black in the body of the paper itself)
- This has been corrected
2 – The interchangeable use of phenotypic (hair / nail color) and genotypic (ee / EE) is problematic. The authors are relying on clinic notes of the breeds and not a genetic analysis of the dogs themselves. And many times the breeds can be aspirational as opposed to true (in other words not all “pure bred” dogs are indeed pure bred). The analyses should be therefore limited to what the authors know (color) and not what is being presumed (genetics). A discussion of possible genetic relationship to the tumor malignancy is not appropriate and necessary at the conclusion, but should be limited to speculation.
- The wording has been slightly changed to “presumed” genetic color, to make sure that the reader understands this is an assumption, rather than a true genetic test. This will be assessed for the development of future studies, where the true genetic color can be verified.
- An addition on the discussion has been made: In this study, no true genetic haircoat analysis was made, but assumption based on the phenotypical color. Nevertheless, this may result in a scaffold for future research studies in which true hair-coat genetic analysis can be performed.
3 – While the authors mention the Broder’s scheme, some orientation for the reader as to how that scheme works relative to the other schemes is warranted. Additionally, a more clear rational for why this scheme is inadequate should be given (other than the rational in human medicine). Has this scheme been proven not to be prognostic? Were these tumors graded by this scheme on the initial pathology report? How did it compare to the IF and TCB schemes?
- To our knowledge, there is no study giving a prognostic correlation with this system in veterinary medicine. These tumors were not graded based on this scheme given that the differentiation of the tumor as an overall (used in this system) was different depending on the area examined, making it too subjective. An addition has been made in the discussion section: “ Broder´s system was not included in this case since, similar to what other studies have shown [11], there can be different grades of differentiation within the same tumor, making it confusing when evaluating the sample. “
4 – For the TCB, how do you rule out that the nests are not 2 dimensional sections through finger like (therefore attached) three dimensional extensions of the tumor? Are you assuming dissociation?
- That is correct, and a mention has been included in the Discussion, regarding limitations of the study: “Additionally, when assessing tumor cell nesting, complete cellular dissociation of the tumor aggregates from the main neoplasia has to be assumed. This can be somewhat problematic given that this is a 2D assessment (a histological slide) of a 3D event, never being sure if the small complexes might be connected to the primary mass in deeper sections or when given a different orientation. This dissociation, however, has to be assumed when assessing the invasive front, given that currently there is no other available system.”
5 – Finally, while the authors describe the few discrepancies between the two schemes, a clear guidance of which is preferred is not given. If the authors believe that a clear preference cannot be made without follow, this should be clearly stated.
- A brief paragraph stating the lack of preference has been included in the discussion: “With this study, conclusions regarding the most accurate grading system for CDSCC cannot be drawn, given that no outcome was available in any case. Nevertheless, given that there are statistical differences between dark and light groups when assessing individual features, one could speculate that in the near future, a new grading system could be elaborated, taking into account these features.”
Smaller areas:
There are several places where the use of language is misleading / confusing. Some examples:
- The title does not make clear there are 2 grading schemes evaluated. Consider: Grading of DSCC in Dogs with light and dark hair by invasive front and tumor cell budding.
- Invasive Front and Tumor Cell Budding Grading Comparison of Digital Squamous Cell carcinoma in Dogs with Dark and Light-haired Color
Line 42: “location, but papilloma virus has not been demonstrated by PCR positivity in affected digits"
- This has been corrected
- This has been corrected None of the photos include size markers or magnification
- We believe that in this case, the magnification and marker size do not give any additional information to demonstrate the features to be highlighted in the picture. If the editor agrees that these changes are really necessary, this can be done. Nevertheless, in this case, more time would be required as new pictures would have to be taken from scratch.
CDSCC and DSCC are used interchangeably. Please make sure that abbreviations are consistent throughout paper.
- This has been addressed and has been made consistent as CDSCC. We wonder if this may sound redundant when using phrases such as “Histological samples of CDSCC from 94 dogs”.
Line 54: Since invasive front is a relatively new concept, a brief orientation such as “invasive front (the area of the margin of the neoplasm consisting of the tumor-host interface)” would be useful to the non-pathologist reader
- A brief explanation has been added: “The invasive front is, as it´s name infers, is the tumor-host interface in which neoplastic cells invade the surrounding stroma, spreading and infiltrating”
Line 83: “genotypic” as mentioned above you are evaluating the visible, phenotypic, hair color only. I suggest removing the (presumed) genetics of the dogs in Table 1 in addition to making the subtypes consistent with the abstract
- We would like to keep the presumed genotype on text and tables (while stating that is “presumed” so it is clear throughout the paper) as there are additional research being currently conducted in which this paper may be use as a scaffold for other publications.
- This has been shortly included in the discussion: “In this study, no true genetic haircoat analysis was made, but assumption based on the phenotypical color. Nevertheless, this may result in a scaffold for future research studies in which true hair-coat genetic analysis can be performed.”
Line 134: Nagamine’s paper specifically does state that the invasive front was evaluated (hence the title of the paper and grading scheme)
- When reading the material and methods, even though the title expressed the evaluation of the invasive front, it was not clear (for us) that all the individual features were assessed only at this point, or if some of the features were taking into account the overall tumor. Nevertheless, this has been taken out. “These features were assessed focusing exclusively on the invasive front. “
Lines 146-148: THANK YOU for providing a standardized area which should be (but is not) specified in all histologic grading scheme!
- Thank you! We also believe this is vital for future grading systems and standardization of methods.
Figures 2-5 – appear twice (although this may be the editors formatting error)
- It is a formatting error but it will be corrected
Line 171: “occasionally obliterating the neoplastic”
- This has been corrected
Figure 4 – use photos of the same magnification
- On figure 4, different magnifications were used to highlight the features of interest. On a lower magnification, nuclear pleomorphism was not obvious on picture B and in close magnification of picture A, the features were not as obvious. Nevertheless, if the editor believes this is really necessary, changes can be made.
Figure 6 – I think the legend was mis-formatted (again, editor error) and that lines 190-192 should be part of the legend and not text
- Formatting error
Lines 197-199: awkward wording. Suggest: “Tumor nests, in contrast, include both these smaller (<5cell) complexes as well as larger aggregates (>15 cells) dissociating . . .” Also, per pint 5 above, consider changing dissociating to “locally separate” or something similar to acknowledge the limitations of 2D histologic sections
- First part has been corrected
- The limitations of the 2D-3D assessment has been included in the discussion. We would rather not change the original wording since that is what is used in previous publications and we I think it makes it more familiar for the reader.. “Additionally, when assessing tumor cell nesting, complete cellular dissociation of the tumor aggregates from the main neoplasia has to be assumed. This can be somewhat problematic given that this is a 2D assessment (a histological slide) of a 3D event, never being sure if the small complexes might be connected to the primary mass in deeper sections or when given a different orientation. This dissociation, however, has to be assumed when assessing the invasive front, given that currently there is no other available system.
Figure 7 – Even in the mitoses and inflammation were not significant, it would be useful to include that information
- An addition has been made within the legend of the table to clarify the reason why these were left out. “Mitoses and inflammation were not included given that they were not statistically significant between groups.”
Line 322 – Wording obscures that these schemes evaluate completely different criteria. “Likewise, the difference criteria of the TCB scheme were also statistically significant between the two populations” (or similar)
- This has been corrected
Line 339 – what color are dogs 74 and 92? So the biggest difference between the schemes was the separation of grades 2 and 3 with a 5 (5,24,25,74,92) going from high to low? Dark more likely to be well differentiated in TCB and light more likely to me poorly differentiated by TCB?
- First portion has been corrected
- I am sorry, I don´t seem to understand the question. Should be the table explained differently maybe?
Lines 360-361 People of Color (capitalize)
- This has been corrected
Discussion: extremely well done. Would just like to see more of a definite answer on preference or lack thereof
- Already assessed according to reviewer 1
Round 2
Reviewer 2 Report
The manuscript is much improved. I only I have few, suggested changes, to the wording and to aid increase the clarity of the tables (which can be done at final formatting). This manuscript does not need to be reviewed again.
Table 1:
The horizontal lines between the groups and text wrapping are slightly off. Editors and authors should make sure the groups are consistent and clear. For example, the complete line above the Labrador Retriever seem to separate that animal from all the dark haired dogs, but based on the numbers, that dog in included in the 11 cited. I assume the line is to separate that dog fromm the genetic color but the Table is not clear. Why is there no hair coat listed for Group 1b (I am assuming black)? Why is there a line to both sides of KB/KB and KB/KY (where the color should go)?
Table 4:
Similar to Table 1, the formatting is slightly off so each category is lined up differently.
When comparing the two different systems, it appears that the Tumor Cell Budding gives a more definitive answer (Grade 1 or 3) as opposed to the Invasive Front System. And that the cases listed at Grade II in the invasive front are categorized as the more concerning Grade 3 by tumor budding. Is this a correct analysis? If so, consider a quick note to that effect. Since the aim is clearly to do further work on these systems looking at prognosis, I suspect that a system which clearly gives an answer will be more likely to clearly show prognosis.
In adding more comments in the discussion, "nevertheless" is used twice in rapid succession in Lines 507 and 510 making the text defensive. Just drop the word from Line 510.
Finally, while I agree that size bars are not necessary for the manuscript, please include in the future as a standard best practices.
